# Preparation, Characterization, and Catalytic Properties of Pd-Graphene Quantum Dot Catalysts

**Jisoo Kim [1], Jeongah Lim [1], Ji Dang Kim [1], Myong Yong Choi [2,\*], Sunwoo Lee [1,\*] and Hyun Chul Choi [1,\*]**

[1] Department of Chemistry, Chonnam National University, Gwangju 61186, Korea; chemistryjisoo@gmail.com (J.K.); 91jeongah@gmail.com (J.L.); chemistry8gee@gmail.com (J.D.K.)

[2] Department of Chemistry and Research Institute of Natural Science, Gyeongsang National University, Jinju 52828, Korea

\* Correspondence: mychoi@gnu.ac.kr (M.Y.C.); sunwoo@chonnam.ac.kr (S.L.); chc12@chonnam.ac.kr (H.C.C.)

**Abstract:** In this study, Pd-graphene quantum dot (Pd-GQD) catalysts were prepared by depositing Pd nanoparticles onto functionalized GQD surfaces, and their morphology and elemental composition were characterized using transmission electron microscopy, X-ray photoelectron spectroscopy, and Raman spectroscopy. The as-prepared Pd-GQD was subsequently employed as a catalyst for the Heck and decarboxylative cross-coupling reactions and was found to exhibit higher catalytic activity than other reference systems. The expanded substrate scope of various substituted aryl iodides further proved that the GQD is an effective support for preparing new heterogeneous catalysts with improved catalytic performances.

**Keywords:** Pd nanoparticle; graphene quantum dot; heterogeneous organocatalysis; Heck cross-coupling reaction; decarboxylative cross-coupling reaction

## 1. Introduction

Pd is a versatile catalyst for the introduction of new carbon–carbon and carbon–heteroatom bonds in molecules [1–4]. Several Pd-catalyzed chemical transformations, such as the Heck, Suzuki, Stille, and Sonogashira cross-coupling reactions, have been successfully developed to synthesize compounds that are used as building blocks in basic organic chemistry research and industrial applications [5–8]. Homogeneous Pd catalysts have usually been used in these coupling reactions due to their high activity and selectivity. However, their practical applications are restricted by several issues, such as the difficult separation of the soluble catalyst from the reaction mixture, the contamination of the final product with toxic residues, and the recycling of the catalyst for consecutive runs [9–11]. In this regard, the development of heterogeneous catalysts, in which the catalyst is immobilized on insoluble solid supports, is gaining extensive attention [12–14]; commonly used supports in this regard include mesoporous silicas [15,16], polymers [17,18], zeolites [19,20], and ionic liquids [21,22]. However, most of the supported catalysts suffer from lower catalytic activity than homogeneous catalysts because of their poor dispersion and leaching from the support. Therefore, the development of new supported Pd catalysts with improved activity and stability remains challenging.

Since the first report in 2008 [23], graphene quantum dots (GQDs) have attracted considerable attention because of their unique and size-dependent properties. They can be produced by cutting nanometric sp$^2$ clusters into graphene sheets, whose lateral size is smaller than 100 nm and whose thickness is less than 10 graphene layers [24–26]. Due to their quantum confinement and surface/edge functional groups, GQDs have the advantages of a size-tunable band gap, stable photoluminescence, and good solubility, which makes them promising candidates for applications in bioimaging, drug delivery, optical display, photovoltaics, and sensors. During the past decade, most research on GQDs has been focused on their synthesis and applications for optical sensing and imaging. Recent



studies have demonstrated that GQDs have great potential for application as catalysts in photocatalytic water splitting, $CO_2$ reduction, and $H_2$ production, which have been well-summarized in several reviews [27–29]. However, the application of catalysts supported on GQDs for organocatalysis has rarely been reported. We have previously reported a novel process for preparing highly dispersed metal or metal oxide nanoparticles supported on $sp^2$ carbon nanostructures, especially carbon nanotubes (CNT) and graphene oxides (GO) [30–33]. The observed results demonstrated that the catalytic performance of Pd catalysts supported on CNT and GO can be significantly improved compared to that of commercial Pd/activated carbon (Pd/C) catalysts in both heterogeneous catalysis and electrocatalysis [34,35].

Our ongoing interest in the development of new effective catalysts, combined with the scarcity of knowledge on the catalytic properties of GQDs in organic reactions, motivated us to investigate the possible use of GQDs as supports in organocatalysis. Herein, we report a new Pd-GQD catalyst, which is prepared by depositing Pd nanoparticles onto the surface of GQDs using N,N'-dicyclohexylcarbodiimide (DCC) as a coupling agent. The morphology and structure of Pd-GQDs were studied using different techniques such as transmission electron microscopy (TEM), X-ray photoelectron spectroscopy (XPS), and Raman spectroscopy. The activity of the Pd-GQD catalysts in Heck and decarboxylative cross-coupling reactions was evaluated, and the results were compared with those obtained using Pd catalysts supported on CNTs and GOs.

## 2. Results

### 2.1. Characterization of the Catalyst

The shape, size distribution, and average size of the GQDs prepared using hydrothermal treatment were determined from the TEM images. Figure 1a shows a low-resolution TEM image of the GQDs, which are indicated by yellow circles. These GQDs exhibit a semi-spherical shape and are dispersed with a narrow size distribution. The corresponding energy-dispersed X-ray (EDX) spectrum shows that the main constituent elements of GQDs are C and O (inset of Figure 1a); no other elements are detected. A histogram of their size distribution (Figure 1b) shows that most of the GQDs are in the range of 3–5 nm, with an average size of ~4.2 nm. The high-resolution TEM image of a single GQD is shown as in the inset of Figure 1b. An interlayer spacing of ~0.29 nm is observed, corresponding to the (100) lattice fringes of a hexagonal graphene structure [36,37]. The lattice spacing of the GQDs is slightly larger than that of pure graphite ($d_{(100)}$ = 0.22 nm), indicating that the presence of functional groups on the surface and edges of graphene sheets can enlarge the basal plane spacing of the GQDs slightly, consistent with previously reported observations on functionalized GQDs [38,39]. The abovementioned results indicate that the GQDs were successfully exfoliated from the parent graphene compound by hydrothermal treatment.

Figure 2 shows TEM images of Pd-GQD, Pd-GO, and Pd-CNT. Figure 2a shows that the GQDs in Pd-GQD were in close contact with the (111) planes of the cubic Pd phase [40,41]. The corresponding EDX spectrum confirms that the Pd nanoparticles were successfully loaded onto the GQD surface (inset of Figure 2a). The observed N signal originates from the amide group on the GQD surface during the DCC-activated process [42]. The adhered Pd nanoparticles could not be separated from the GQDs, even after thorough washing and prolonged sonication. Such strong adhesion between the GQDs and Pd nanoparticles possibly results from the relatively high binding energy between Pd and the surface functional groups on the GQDs or the formation of a multijunction induced by coupling to each other. As shown in Figure 2b, the paper-like structure of the GO supported the nanoparticles, which were clearly visible as dark spots in the TEM image. For the Pd-CNTs, the nanoparticles are highly dispersed along the entire CNT walls (Figure 2c). Most of the nanoparticles in both samples were 1–3 nm in size. The corresponding EDX spectra of the Pd-GOs and Pd-CNTs reveal that the nanoparticles on both carbon supports consist of Pd (insets of Figure 2b,c). The observed Cu signal originates from the TEM grid, and no other peaks characteristic of impurities were observed. The Pd contents of Pd-GQD, Pd-GO,

and Pd-CNT were evaluated using the XPS spectrum (Figure 2d). The relative surface atomic ratios were estimated from the corresponding peak areas and were corrected using atomic sensitivity factors. The calculated Pd contents of Pd-GQD, Pd-GO, and Pd-CNT were approximately 2.8%, 8.3%, and 4.3%, respectively. To check the bulk concentration, the Pd content in the samples was determined by inductively coupled plasma atomic emission spectroscopy (ICP-AES). The determined Pd contents of Pd-GQD, Pd-GO, and Pd-CNT were 3.1%, 8.7% and 4.9%, respectively, which was consistent with that of the XPS results.

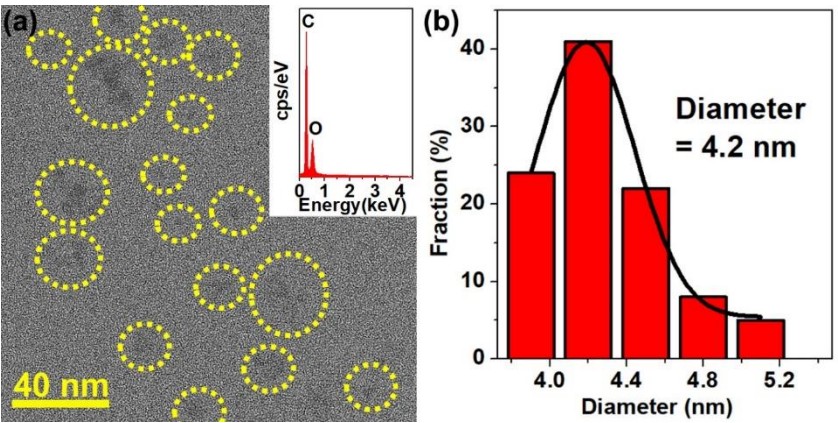

**Figure 1.** (**a**) TEM image of GQDs at low magnification. The inset shows the corresponding EDX spectrum. (**b**) The size-distribution histogram of GQDs. The solid line in the histogram represents a Gaussian fitting curve.

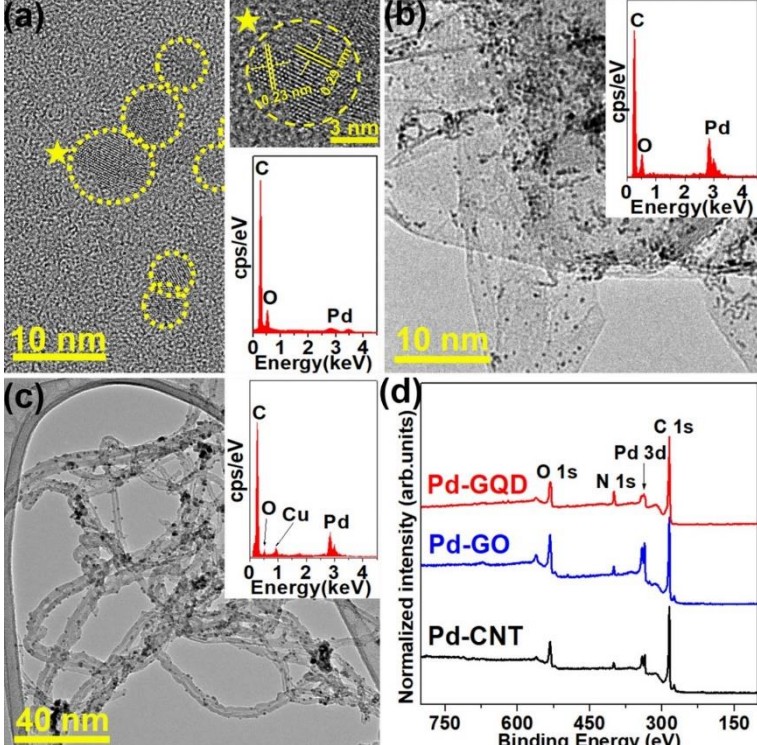

**Figure 2.** (**a**) TEM image of Pd-GQDs at low magnification. The inset (**top**) shows the enlarged TEM image of the marked area in (**a**). The inset (**bottom**) provides the corresponding EDX spectrum. (**b**) TEM image of Pd-GO with its corresponding EDX spectrum (inset). (**c**) TEM image of Pd-CNT with its corresponding EDX spectrum (inset). (**d**) X-ray photoelectron spectroscopy (XPS) survey results of Pd-GQD, Pd-GO, and Pd-CNT.

Raman spectroscopy was used to investigate the degree of deformation in the graphite structure of nanostructured carbon supports during the preparation process. The Raman spectra of all samples, viz., pristine GQD, Pd-GQD, Pd-GO, and Pd-CNT (Figure 3), showed two characteristic D and G bands centered at ~1340 and ~1580 $cm^{-1}$, respectively. The D band is related to the breathing mode of $sp^3$ carbons in defective areas of the carbon network, whereas the G band is assigned to the stretching mode of $sp^2$ carbons in the graphite plane [43,44]. The relative intensity ratio of the D and G bands ($I_D/I_G$) is used to estimate the degree of crystallinity in carbon nanostructures. A small $I_D/I_G$ value indicates that the carbon nanostructures contain more crystalline $sp^2$ than disordered $sp^3$ domains induced by the defects. The calculated $I_D/I_G$ values for pristine GQD, Pd-GQD, Pd-GO, and Pd-CNT were approximately 1.05, 1.15, 1.57, and 1.52, respectively, indicating that the GQDs maintained the intrinsic properties of the graphite structure compared to the GOs and CNTs during the preparation process. The observed $I_D/I_G$ value of Pd-GQD is comparable to those for previously reported GQDs [36,45,46].

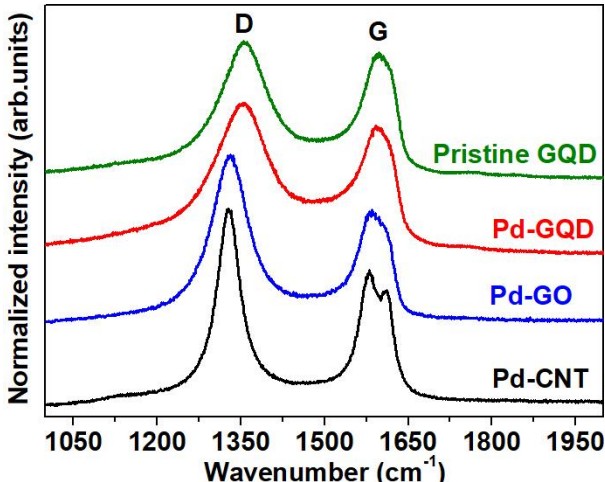

**Figure 3.** Raman spectra of pristine GQD, Pd-GQD, Pd-GO, and Pd-CNT.

*2.2. Organic Catalysis Study*

The catalytic activities of the different Pd catalysts were first evaluated in the Heck cross-coupling reaction. 4-Iodotoluene (**1a**) and n-butyl acrylate (**2**) were chosen as standard substrates and were reacted in the presence of Pd catalysts and $K_3PO_4$ at 85 °C for 3 h. Among the catalysts tested, Pd-GQD showed the highest yield in the Heck cross-coupling reaction. As shown in Table 1, using Pd-GQD afforded the desired Heck cross-coupled product **3a** in 84% yield (entry 1), while the use of Pd-GO afforded **3a** in 72% yield (entry 2). When Pd-CNT was used, **3a** was formed in 49% yield (entry 3). However, the use of commercially available Pd/C, which is widely employed in organic reactions, resulted in the formation of **3a** in only 15% yield (entry 4). The reactions with only the QGD, GO, and CNT supports (without Pd) did not afford the desired product (entries 5–7). As expected, no product was formed in the absence of the Pd catalyst (entry 8). To expand the substrate scope of the reaction, several attempts were made using various substituted aryl iodides with n-butyl acrylate in the presence of Pd-GQD, and the obtained results are summarized in Scheme 1. The aryl iodides bearing electron-donating groups such as ethyl and methoxy on the aryl group gave **3b** and **3c** in 80% and 75% yields, respectively. On the other hand, aryl iodides bearing halogen groups reacted with n-butyl acrylate to afford the corresponding products **3d**, **3e**, and **3f** in 88%, 87%, and 85% yields, respectively. 4-Iodoacetophenone, which is an aryl iodide with a ketone group, afforded the desired product **3g** in 92% yield.

**Table 1.** Catalytic activities in the Pd-catalyzed Heck cross-coupling reactions.

| Entry | Pd Catalyst | Yield (%) |
|-------|-------------|-----------|
| 1 | Pd-GQD | 84 |
| 2 | Pd-GO | 72 |
| 3 | Pd-CNT | 49 |
| 4 | Commercial Pd/C | 15 |
| 5 | GQD | Not detected |
| 6 | GO | Not detected |
| 7 | CNT | Not detected |
| 8 | - | Not detected |

Reaction conditions: **1a** (0.3 mmol), **2** (0.15 mmol), Pd catalyst (3.5 mol% based on Pd) and $K_3PO_4$ (0.6 mmol) were reacted in DMF (1.2 mL) at 85 °C for 3 h. Yields are determined by gas chromatography with internal standard.

**Scheme 1.** Cross-coupling reactions of substituted aryl iodides with n-butyl acrylate.

Next, we focused on the decarboxylative coupling reaction of alkynoic acids and aryl iodides. Decarboxylative coupling reactions using alkynoic acids derivatives have been widely used as an alternative to Sonogashira cross-coupling reactions [47]. Different Pd catalysts were tested in the reaction between 4-iodotoluene and phenylpropiolic acid in the presence of TBAF in DMSO at 80 °C for 24 h, and the results are listed in Table 2. The employment of Pd-GQD as a catalyst afforded the decarboxylative coupled product **5a** in 82% yield (entry 1). On the other hand, the use of Pd-GO, Pd-CNT, and Pd/C afforded **5a** in 41%, 35%, and 34% yields, respectively (entries 2–4). No product was formed when only GQD was employed as the catalyst or in the absence of a Pd catalyst (entries 5 and 6). These results suggested that Pd-GQD was the most effective catalyst for the decarboxylative coupling reaction. To broaden the substrate scope, various aryl iodides were reacted with phenylpropiolic acid, the results of which are summarized in Scheme 2. 4-Iodoanisole gave

**5b** in a 71% yield. 4-Biphenyliodide and 2-iodonaphthalene produced the corresponding products **5c** and **5d** in 78% and 71% yields, respectively. Aryl iodides bearing halide groups such as bromide, chloride, and fluoride afforded the desired products **5e**, **5f**, and **5g** in good yields.

**Table 2.** Catalytic activities in the Pd-catalyzed decarboxylative cross-coupling reactions.

| Entry | Pd Catalyst | Yield (%) |
|-------|-------------|-----------|
| 1 | Pd-GQD | 82 |
| 2 | Pd-GO | 41 |
| 3 | Pd-CNT | 35 |
| 4 | Commercial Pd/C | 34 |
| 5 | GQD | Not detected |
| 6 | - | Not detected |

Reaction conditions: **1a** (0.3 mmol), **4** (0.15 mmol), Pd catalyst (3.5 mol% based on Pd) and TBAF (0.45 mmol) were reacted in DMSO (1.2 mL) at 80 °C for 24 h. Yields are determined by gas chromatography with internal standard.

**Scheme 2.** Cross-coupling reactions of substituted aryl iodides with phenylpropiolic acid.

These results clearly indicated the beneficial effects of GQDs on heterogeneous catalytic reactions compared to other carbon supports. The excellent activity of Pd-GQDs is attributed to the functionalized GQD support. The presence of GQDs enables spatial confinement of Pd nanoparticles without aggregation and consequently provides the high surface area of the Pd catalyst. In addition, the reactant–product mass transportation is enhanced by $\pi$–$\pi$ interactions between the functional groups on the GQD surface and the aromatic rings of the reactants. Further XPS studies are in progress to determine the surface properties of the Pd-GQDs and other supports to better understand the role of these supports in catalytic performance.

## 3. Materials and Methods

### 3.1. Materials

Palladium(II) sodium chloride ($Na_2PdCl_4$), sodium hydrosulfide hydrate ($NaSH \cdot xH_2O$), DCC, ethylenediamine, natural graphite powder, and commercial Pd/C (10 wt.% Pd) were

purchased from Sigma-Aldrich. The CNTs were obtained from Carbon Nano Tech. Co., Ltd. (South Korea). The GOs were prepared from natural graphite powder using the modified Hummers method [48]. The experimental details can be found in our previous report [31]. All other reagents and solvents were purchased from Sigma-Aldrich and were used without additional purification. All aqueous solutions were prepared using deionized (DI) water from a Direct Q3 Millipore system.

### 3.2. Preparation of GQDs

The GQDs were prepared from natural graphite powder using a previously reported one-step hydrothermal method [49,50]. The graphite powder (1.0 g) was added to concentrated $H_2SO_4$ (100 mL) to prepare a suspension, to which $NaNO_3$ (43.0 g) was added with vigorous stirring. The resulting mixture was cooled to 0 °C in an ice bath. Then, $KMnO_4$ (3.0 g) was gradually added with stirring and cooling to ensure that the temperature of the mixture did not exceed 20 °C. The mixture was stirred at 40 °C for 1 h and heated to 120 °C for 12 h. After being cooled to room temperature, 500 mL of DI water was added to the mixture. The mixture was then transferred to an autoclave and heated to 180 °C for 12 h. The resulting precipitate was collected by repeated filtering and washing with pure ethanol and DI water. Finally, the GQD powders were obtained by drying at 60 °C under vacuum overnight.

### 3.3. Preparation of Pd-GQDs

Pd catalysts supported on GQDs, GOs, or CNTs were prepared following a previously reported DCC-activated method [30]. Briefly, 0.4 g GQD was loaded in a 250 mL round bottom flask, and 25 mL THF was added under sonication for 20 min. Then, 50 mL of a 1:1 DCC/THF mixture was poured into the flask under sonication for 30 min. The resulting suspension was filtered and washed with THF and $CH_3OH$. The isolated solid (~3 mg/mL) was re-dispersed in THF, and an ethylenediamine/THF (~2 mL/mL) solution was added with continuous stirring for 1 h. Finally, 0.2 mg of $Na_2PdCl_4$ was dissolved in 10 mL DI water and stirred. An ice-cooled 0.1 M $NaBH_4$ solution was added in one step to the solution with continuous stirring. After the reduction, 20 mg of DCC-activated GQD powder was added to the solution in one step. The precipitate was collected by repeated filtering and washing with pure ethanol and DI water. Finally, the Pd-GQD catalyst was obtained by drying at 60 °C under vacuum overnight. The loading of Pd onto the GQDs, GOs, or CNTs was calculated from the XPS spectra without further analysis.

### 3.4. Characterization

TEM and EDX analyses were performed on an FEI Tecnai-F20 microscope (Philips, Netherlands) operated at an acceleration voltage of 200 kV. Samples for TEM imaging were prepared by depositing a colloidal suspension of Pd-GQD on a Cu grid. Raman spectra were recorded at room temperature on a Renishaw 1000 micro-Raman spectrometer (Renishaw, UK) using an argon ion laser ($\lambda$ = 514.5 nm). XPS analysis was performed using a VG Multilab 2000 spectrometer (ThermoVG Scientific, UK) equipped with an MgK$\alpha$ source (1253.6 eV) and a charge neutralizer. For analyzing the XPS peaks, the C 1s peak position was set as 284.5 eV and was used as the internal reference to locate the other peaks. The content of Pd in the sample was determined by ICP-AES with an OPTIMA 4300 DV (Perkin Elmer). Prior to the measurement, the sample was treated with a mixture of $HBO_3$, HF, and $HNO_3$ in order to dissolve it completely.

### 3.5. Catalytic Activity Test

The catalytic activities of the different Pd catalysts were tested for Heck and decarboxylative cross-coupling reactions. Since the different samples have different Pd contents, the catalytic evaluations were conducted in a reaction vial using 3.5 mol% of the Pd catalyst based on the Pd content of the sample. For the Heck reaction, the vial was charged with certain amounts of aryl iodide, n-butyl acrylate, Pd catalyst, and $K_3PO_4$ in DMF, and

the reaction was conducted at 85 °C for 3 h. 4-Iodotoluene was selected as the model compound to establish optimal reaction conditions. For the Pd-catalyzed decarboxylative reaction, 4-iodotoluene, phenylpropiolic acid, Pd catalyst, and TBAF in DMSO were placed in a reaction vial and were reacted at 80 °C for 24 h. To broaden the substrate scope of the reaction system, we studied several additional reactions between various substituted aryl iodides and phenylpropiolic acid in the presence of a Pd catalyst. The progress of the reaction was monitored by thin layer chromatography, and conversion of the product was determined by GC analysis using naphthalene as the internal standard.

## 4. Conclusions

The enhancement of catalytic activity is one of the most consistently pursued research topics in the field of heterogeneous catalysis. In this study, we have successfully prepared a heterogenous Pd-GQD catalyst and used it to catalyze the Heck and decarboxylative cross-coupling reactions. This is the first reported example in literature that illustrates the use of GQD as a support in organic catalysis. Pd-GQD shows higher catalytic activity for these coupling reactions than reference systems, such as Pd-GO, Pd-CNT, and commercial Pd/C catalyst. The enhanced activity of Pd-GQD can be attributed to the increased dispersion of Pd nanoparticles or favorable reactant–product mass transportation induced by the functionalized GQDs. Therefore, GQD is confirmed to be an effective support for preparing new heterogeneous metal/metal oxide catalysts with improved performances. Further investigations of the surface properties and long-term stability of this material is in progress.

**Author Contributions:** Conceptualization and methodology, H.C.C.; investigation, J.K., J.L. and J.D.K.; supervision, M.Y.C., S.L. and H.C.C.; writing—original draft, H.C.C.; writing—review and editing, H.C.C.; funding acquisition, H.C.C. All authors have read and agreed to the published version of the manuscript.

**Funding:** This work was supported by the National Research Foundation of Korea (NRF) grant funded by the Korea government (MSIT) grant number 2021R1F1A1061918.

**Acknowledgments:** The authors are grateful to the Center for Research Facilities at the Chonnam National University for their assistance in the XPS analysis. The authors are also thankful to the Korea Basic Science Institute–Gwangju branch for their assistance in the TEM and Raman analyses. The authors also acknowledge the Core-Facility Center for Photochemistry and Nanomaterials of Gyeongsang National University.

**Conflicts of Interest:** The authors declare no conflict of interest.

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
