# Peer review of "Preparation, Characterization, and Catalytic Properties of Pd-Graphene Quantum Dot Catalysts"

_catalysts, doi:10.3390/catal12060619_

Round 1

Reviewer 1 Report

Line 97. “Most of the nanoparticles in both samples were 3–6 nm in size”. Looking at the scale bar in Figures 2b and 2c, most of the nanoparticles appear to be less than 1 nm in size.

Line 108. “The inset shows a high-resolution TEM image of the single GQD.” There is no inset there.

Author Response

Responses to Reviewer #1:

Line 97. “Most of the nanoparticles in both samples were 3–6 nm in size”. Looking at the scale bar in Figures 2b and 2c, most of the nanoparticles appear to be less than 1 nm in size.

→ Authors’ response: Thanks to the reviewer for the comment. According to your suggestion, we revised the sentence at line 97 as follows:

“Most of the nanoparticles in both samples were 1–3 nm in size.”

Line 108. “The inset shows a high-resolution TEM image of the single GQD.” There is no inset there.

→ Authors’ response: Thanks to the reviewer for the comment. That was a writing error on our part. We have deleted the sentence at line 112 in the revised manuscript.

Reviewer 2 Report

This is a neat study wherein authors have synthesized Pd-graphene quantum dots succesfully and applied them for coupling reactions illustrating superior performances by Pd GQDs when comparing them to carbon nanotubes and graphene oxide. While overall this study will be unambigously of interest to the readers of this journal, technical soundness can be improved, vastly in some aspects. 

1. Firstly, please provide Pd loading using ICP-based techniques which leads to far more accurate determination of bulk Pd concentrations on such systems than XPS. 

2. Authors approach of structurally characterizing GQDs using Raman and illustrating that the band intensity is most favorable for SP2 graphitic carbons than saturated SP3 carbon atoms is reasonable however, authors should consider including unfunctionalized GQDs for completeness (including it in SI or a discussion would suffice), as to ascertain if Raman behavior of these interesting materials change to some extent post Pd-functionalization.  

3. It is not unlikely that Pd sites supported on GQDs are optimally located for both types of chemistries. Afterall, Pd loading amount is lowest for the Pd-GQD sample and authors have experimentally normalized the Pd content by using 3.5 mol% w.r.t substrate for the two types of reactions. However, since average diameters of Pd in the three distinct samples may be vastly different, to the extent that superior catalytic performance is just due to higher fraction of exposed Pd surfaces in GQDs, authors should discuss size dependent catalytic differences in more detail, align it with their findings, and include some of recent examples from literature also.

Overall, this submission is publishable in this journal provided the points outlined above have been addressed. 

Author Response

Responses to Reviewer #2:

This is a neat study wherein authors have synthesized Pd-graphene quantum dots successfully and applied them for coupling reactions illustrating superior performances by Pd GQDs when comparing them to carbon nanotubes and graphene oxide. While overall this study will be unambiguously of interest to the readers of this journal, technical soundness can be improved, vastly in some aspects. 

→ The authors are grateful for the positive feedback of the submitted investigation and useful comments.

  1. Firstly, please provide Pd loading using ICP-based techniques which leads to far more accurate determination of bulk Pd concentrations on such systems than XPS. 

→ Authors’ response: Thanks to the reviewer for the comment. According to your suggestion, the ICP-AES data of Pd-GQD, Pd-GO, and Pd-CNT was added in the revised manuscript. The ICP-AES results are added new sentences at lines 104-108 as follows:

“To check the bulk concentration, the Pd content in the samples was determined by inductively coupled plasma atomic emission spectroscopy (ICP-AES). The determined Pd contents of Pd-GQD, Pd-GO, and Pd-CNT were 3.1%, 8.7% and 4.9%, which was consistent with that of the XPS results.”

And, we inserted detailed descriptions of ICP-AES measurement at lines 234-236 as follows:

The content of Pd in the sample was determined by ICP-AES with an OPTIMA 4300 DV (Perkin Elmer). Prior to the measurement, the sample was treated with a mixture of HBO3, HF and HNO3 in order to dissolve it completely.”

  1. Authors approach of structurally characterizing GQDs using Raman and illustrating that the band intensity is most favorable for SP2 graphitic carbons than saturated SP3 carbon atoms is reasonable however, authors should consider including unfunctionalized GQDs for completeness (including it in SI or a discussion would suffice), as to ascertain if Raman behavior of these interesting materials change to some extent post Pd-functionalization.  

→ Authors’ response: Thanks to the reviewer for the comment. According to your suggestion, the Raman spectrum of pristine GQDs was added in the Figure 3.

The Raman results of pristine GQDs are added in paragraph at lines 104-108 as follows:

“The Raman spectra of all samples, viz. pristine GQD, Pd-GQD, Pd-GO, and Pd-CNT (Figure 3) showed two characteristic D and G bands centered at ~1340 and ~1580 cm−1, respectively. The D band is related to the breathing mode of sp3 carbons in defective areas of the carbon network, whereas the G band is assigned to the stretching mode of sp2 carbons in the graphite plane [43, 44]. The relative intensity ratio of the D and G bands (ID/IG) is used to estimate the degree of crystallinity in carbon nanostructures. A small ID/IG value indicates that the carbon nanostructures contain more crystalline sp2 than disordered sp3 domains induced by the defects. The calculated ID/IG values for pristine GQD, Pd-GQD, Pd-GO, and Pd-CNT were approximately 1.05, 1.15, 1.57, and 1.52, respectively, indicating that the GQDs maintained the intrinsic properties of the graphite structure compared to the GOs and CNTs during the preparation process. The observed ID/IG value of Pd-GQD is comparable to those for previously reported GQDs [36, 45, 46].”

And, we revised caption of Figure 3 at line 135 as follows:

Figure 3. Raman spectra of pristine GQD, Pd-GQD, Pd-GO, and Pd-CNT.”

  1. It is not unlikely that Pd sites supported on GQDs are optimally located for both types of chemistries. Afterall, Pd loading amount is lowest for the Pd-GQD sample and authors have experimentally normalized the Pd content by using 3.5 mol% w.r.t substrate for the two types of reactions. However, since average diameters of Pd in the three distinct samples may be vastly different, to the extent that superior catalytic performance is just due to higher fraction of exposed Pd surfaces in GQDs, authors should discuss size dependent catalytic differences in more detail, align it with their findings, and include some of recent examples from literature also.

→ Authors’ response: We thank reviewer for this important comment. The performance of heterogeneous catalysts is governed by various factors such as their size, shape, composition, degree of dispersion on supports, and surface properties. We also examined the size effect on the catalytic activity using various sp2 carbon supports. However, we did not get the direct relationship between the particle size and the activity in the three distinct samples. Therefore, we did not include any experimental results about that in this manuscript. The present study only focused on the possible use of GQDs as supports in organocatalysis. We will consider the effect of particle size and surface structure on the catalytic activity in the future works.

Overall, this submission is publishable in this journal provided the points outlined above have been addressed. 

Finally, we added the new acknowledgment for the ICP-AES measurement at lines 273-274 as follows:

“The authors also acknowledge the Core-Facility Center for Photochemistry & Nanomaterials of Gyeongsang National University.”
